# Open-source Large Language Models are Strong Zero-shot Query Likelihood Models for Document Ranking

**Shengyao Zhuang**[1], **Bing Liu**[1], **Bevan Koopman**[1,2], **Guido Zuccon**[2]
[1]CSIRO, [2]The University of Queensland
[1]{shengyao.zhuang,bing.liu,bevan.koopman}@csiro.au
[2]g.zuccon@uq.edu.au

## Abstract

In the field of information retrieval, Query Likelihood Models (QLMs) rank documents based on the probability of generating the query given the content of a document. Recently, advanced large language models (LLMs) have emerged as effective QLMs, showcasing promising ranking capabilities. This paper focuses on investigating the genuine zero-shot ranking effectiveness of recent LLMs, which are solely pre-trained on unstructured text data without supervised instruction fine-tuning. Our findings reveal the robust zero-shot ranking ability of such LLMs, highlighting that additional instruction fine-tuning may hinder effectiveness unless a question generation task is present in the fine-tuning dataset. Furthermore, we introduce a novel state-of-the-art ranking system that integrates LLM-based QLMs with a hybrid zero-shot retriever, demonstrating exceptional effectiveness in both zero-shot and few-shot scenarios. We make our codebase publicly available at https://github.com/ielab/llm-qlm.

## 1 Introduction

Ranking models (or rankers) are a fundamental component in many information retrieval (IR) pipelines. Pre-trained language models (PLMs) have recently been leveraged across bi-encoder (Karpukhin et al., 2020; Xiong et al., 2021; Zhuang and Zuccon, 2021c; Wang et al., 2022a; Gao and Callan, 2022), cross-encoder (Nogueira and Cho, 2019; Nogueira et al., 2020; Zhuang et al., 2021), and sparse (Lin and Ma, 2021; Formal et al., 2021; Zhuang and Zuccon, 2021b) ranker architectures, showing impressive ranking effectiveness.

Despite this success, the strong effectiveness of PLM-based rankers does not always generalise without sufficient in-domain training data (Thakur et al., 2021; Zhuang and Zuccon, 2021a, 2022). Transferring knowledge from other domains has been used to overcome this issue (Lin et al., 2023) by training these rankers on large-scale supervised QA datasets such as MS MARCO (Nguyen et al., 2017). Alternatively, generative large language models (LLMs) like GPT3 (Brown et al., 2020) have been used to synthesize domain-specific training queries, which are then used to train these rankers (Bonifacio et al., 2022; Dai et al., 2023). Despite their effectiveness, all of these methods consume significant expenses in training a PLM-based ranker.

In this paper, we consider a third avenue to address this challenge: leveraging LLMs to function as Query Likelihood Models (QLMs) (Ponte and Croft, 1998; Hiemstra, 2000; Zhai and Lafferty, 2001). Essentially, QLMs are expected to understand the semantics of documents and queries, and estimate the possibility that each document can answer a certain query. Notably, recent advances in this direction have greatly enhanced the ranking effectiveness of QLM-based rankers by leveraging PLMs like BERT (Devlin et al., 2019) and T5 (Raffel et al., 2020). These PLM-based QLMs are fine-tuned on query generation tasks and subsequently employed to rank documents as per their likelihood (Nogueira dos Santos et al., 2020; Zhuang et al., 2021; Lesota et al., 2021; Zhuang and Zuccon, 2021c).

We focus on a specific PLM-based QLM, the recently proposed *Unsupervised Passage Re-ranker* (UPR) (Sachan et al., 2022). UPR leverages advanced LLMs to obtained the query likelihood estimations. Empirical results show that using the T0 LLM (Sanh et al., 2022) as a QLM, large gains in ranking effectiveness can be obtained. A key aspect of this work is that this effectiveness is obtained without requiring additional fine-tuning data, making Sachan et al. highlight the zero-shot ranking capabilities of their LLM-based QLM. However, we argue that the experimental setting used by Sachan et al. does not fully align with a *genuine zero-shot* scenario for the QLM ranking task. This is because T0 has already undergone fine-tuning on numer-

ous question generation (QG) tasks and datasets, subsequent to its unsupervised pre-training.[1] Consequently, there exists a discernible task leakage to the downstream QLM ranking task, thereby rendering their approach more akin to a transfer learning setting, rather than a true zero-shot approach.

To gain a comprehensive understanding of the zero-shot ranking capabilities of LLM-based QLM rankers, in this paper we take a fresh examination of this topic. Our approach involves harnessing the power of state-of-the-art transformer decoder-only LLMs, such as LLaMA (Touvron et al., 2023), which have undergone pre-training solely on unstructured text through unsupervised next token prediction. Importantly, the models we consider have not undergone any additional supervised instruction fine-tuning, ensuring a truly complete zero-shot setting for our investigation.

We further extend our analysis by comparing the effectiveness of these LLMs with various popular instruction-tuned LLMs in the context of zero-shot ranking tasks. Interestingly, our findings reveal that further instruction fine-tuning adversely affects the effectiveness of QLM ranking, particularly when the fine-tuning datasets lack specific QG tasks. This insight highlights the strong zero-shot QLM ranking ability of LLMs that solely rely on pre-training, thereby suggesting that further instruction fine-tuning is unnecessary for achieving strong zero-shot effectiveness. Building upon these insights, we push the boundaries of zero-shot ranking even further by integrating a hybrid zero-shot first-stage retrieval system, followed by re-ranking using the zero-shot LLM-based QLM re-rankers and a relevance score interpolation technique (Wang et al., 2021). Our approach achieves state-of-the-art effectiveness in zero-shot ranking on a subset of the BIER dataset (Thakur et al., 2021).

## 2 Methodology

**Zero-shot QLM re-ranker:** We follow the setting introduced in previous works (Zhuang et al., 2021; Sachan et al., 2022) to evaluate the zero-shot QLM ranking capability of LLMs. Specifically, given a sequence of query tokens $q$ and a set $D$ containing candidate documents retrieved by a first-stage zero-shot retriever such as BM25, the objective is to rank all candidate documents $d \in D$ based on

the average log likelihood of generating all query tokens, as estimated by a LLM. The relevance scoring function is defined as:

$$S_{QLM}(\boldsymbol{q}, d) = \frac{1}{|\boldsymbol{q}|} \sum_t \log \mathrm{LLM}(q_t | \boldsymbol{p}, d, \boldsymbol{q}_{<t}) \quad (1)$$

here, $q_t$ denotes the $t$-th token of the query, $\boldsymbol{p}$ is a model and task specific prompt used for prompting the LLM to behave like a question generator (see Appendix A for more details), and $\mathrm{LLM}(q_t | \boldsymbol{p}, d, \boldsymbol{q}_{<t})$ refers to the probability of generating the token $q_t$ given the prompt $\boldsymbol{p}$, the candidate document $d$, and the preceding query tokens $\boldsymbol{q}_{<t}$. It is important to note that, in a truly zero-shot ranking pipeline, the first-stage retriever should be a zero-shot method and the QLM re-ranker should exclusively be pre-trained on unsupervised unstructured text data and no fine-tuning is performed using any QG data.

**Interpolating with first-stage retriever:** Following Wang et al. (2021), instead of solely relaying on the query likelihood scores estimated by the LLMs, we also linearly interpolate the QLM score with the BM25 scores from the first-stage retriever by using the weighted score sum:

$$S(\boldsymbol{q}, d) = \alpha \cdot S_{BM25}(\boldsymbol{q}, d) + (1 - \alpha) \cdot S_{QLM}(\boldsymbol{q}, d), \quad (2)$$

Here, $\alpha \in [0, 1]$ represents the weight assigned to balance the contribution of the BM25 score and the QLM score. In our experiments, we heuristically apply min-max normalization to the scores and assign more weight to the QLM scores, given its pivotal role as the second-stage re-ranker. This is achieved by setting $\alpha = 0.2$ without conducting any grid search. We use the python library ranx[2] (Bassani and Romelli, 2022) to implement the interpolation algorithm.

**Few-shot QLM re-ranker:** Since LLMs are strong few-shot learners (Brown et al., 2020), we also conducted experiments to explore how LLM-based QLM re-rankers could be further enhanced by providing a minimal number of human-judged examples. To achieve this, we employed a prompt template known as "Guided by Bad Questions" (GBQ) (Bonifacio et al., 2022). The GBQ template consists of only three document, good question, and bad question triples. We use it to guide the LLM-based QLM to produce more accurate query likelihood estimations. We refer readers to the original paper for details about the GBQ template.

---

[1]There are at least 16 QG datasets according to the open-sourced T0 training: https://huggingface.co/datasets/bigscience/P3

[2]https://github.com/AmenRa/ranx

# 3 Experimental Settings

**LLMs:** Our focus is on the response of LLMs in the QLM ranking task, specifically in a genuine zero-shot setting. To accomplish this, we used LLaMA (Touvron et al., 2023) and Falcon (Almazrouei et al., 2023), both of which are transformer decoder-only models that are pre-trained solely on large, publicly available unstructured datasets (Penedo et al., 2023). We specifically consider open-source LLMs because we can control the data used to train them, thus guaranteeing no QG dataset was used.

To evaluate the influence of instruction fine-tuning data on QLM estimation, we compared these models with other well-known LLMs that were fine-tuned with instructions, including T5 (Raffel et al., 2020), Alpaca (Taori et al., 2023), StableLM, StableVicuna, and Falcon-instruct (Almazrouei et al., 2023). It is important to note that the fine-tuning instruction data for these models are unlikely to include QG tasks.[3] Additionally, we follow Sachan et al. (2022) to include T0 (Sanh et al., 2022) and FlanT5 (Chung et al., 2022), which underwent fine-tuning specifically for QG instructions. All LLMs used in this paper are openly available, see Appendix B for more details.

**Baselines and datasets:** In our evaluation, we compared LLM-based QLMs with several existing methods, including BM25 (Robertson and Zaragoza, 2009), QLM-Dirichlet (Zhai and Lafferty, 2004), Contriever (Izacard et al., 2022), and HyDE (Gao et al., 2022), which are zero-shot first-stage retrievers. We also compared them with fine-tuned retrievers and re-rankers trained on MS MARCO passage ranking data, representing a transfer learning setting. Specifically, the evaluated retrievers are Contriever-msmarco, SPLADE-distill (Formal et al., 2022), and DRAGON+ (Lin et al., 2023), while the re-rankers are T5-QLM-large (Zhuang et al., 2021), monoT5-3B (Nogueira et al., 2020), and monoT5-3B-Inpars-v2 (Jeronymo et al., 2023). Additionally, we compared our best QLM ranking pipeline with PROMPTAGATOR (Dai et al., 2023), a state-of-the-art zero-shot and few-shot method. See Appendix C for detailed information about the baselines.

To ensure feasibility, we conducted experiments on a popular subset of the BEIR benchmark

---

Table 1: Main results. Re-rankers re-rank Top100 documents retrieved by BM25. Transferred retrievers and re-rankers are fine-tuned on MS MARCO.

| Methods | TRECC | DBpedia | FiQA | Robust04 | Avg |
|---|---|---|---|---|---|
| **Zero-shot Retrievers** | | | | | |
| BM25 | 59.5 | 31.8 | 23.6 | 40.7 | 38.9 |
| QLM-Dirichlet | 50.8 | 29.5 | 20.5 | 40.7 | 35.4 |
| Contriever | 23.3 | 29.2 | 24.5 | 31.6 | 27.2 |
| HyDE | 58.2 | 37.2 | 26.6 | 41.8 | 41.0 |
| **Instruction tuned QLM Re-rankers** | | | | | |
| **Without QG task** | | | | | |
| T5-3B | 48.7 | 21.9 | 16.2 | 38.0 | 31.2 |
| T5-11B | 67.9 | 33.7 | 31.0 | 27.4 | 40.3 |
| Alpaca-7B | 67.1 | 35.0 | 33.7 | 44.6 | 45.1 |
| StableLM-7B | 74.0 | 37.2 | 34.1 | 48.3 | 48.4 |
| StableVicuna-13B | 71.8 | 39.4 | 39.1 | 51.3 | 50.4 |
| Falcon-7B-instruct | 66.8 | 38.2 | 33.4 | 50.7 | 47.3 |
| Falcon-40B-instruct | 70.2 | 40.5 | 40.9 | 51.3 | 50.7 |
| **With QG task** | | | | | |
| T0-3B | 71.6 | 38.8 | 41.4 | 50.1 | 50.5 |
| T0-11B | 73.9 | 38.7 | **43.8** | 49.7 | 51.5 |
| FlanT5-3B | 71.1 | 39.7 | 41.2 | 50.0 | 50.5 |
| FlanT5-11B | 74.9 | **41.7** | 43.3 | 52.4 | **53.1** |
| **Zero-shot QLM Re-rankers** | | | | | |
| LLaMA-7B | 69.4 | 39.9 | 41.5 | 53.6 | 51.1 |
| LLaMA-13B | 69.8 | 37.6 | 41.8 | **54.2** | 50.9 |
| Falcon-7B | 73.3 | **41.7** | 41.3 | 52.5 | 52.2 |
| Falcon-40B | **75.2** | 41.0 | 43.1 | 53.1 | **53.1** |
| **Transferred Retrievers** | | | | | |
| Contriever-msmarco | 59.6 | 41.3 | 32.9 | 47.3 | 45.3 |
| SPLADE-distill | 71.1 | 44.2 | 35.1 | 45.8 | 49.1 |
| DRAGON+ | 75.9 | 41.7 | 35.6 | 47.9 | 50.3 |
| **Transferred Re-rankers** | | | | | |
| T5-QLM-large | 71.4 | 38.0 | 39.0 | 47.7 | 49.0 |
| monoT5-3B | 79.8 | 44.8 | 46.0 | 56.2 | 56.7 |
| monoT5-3B-InPars-v2 | 83.8 | 46.6 | 46.1 | 58.5 | 58.8 |

---

datasets[4]: TRECC (Voorhees et al., 2021), DBPedia (Hasibi et al., 2017), FiQA (Maia et al., 2018), and Robust04 (Voorhees, 2005). The evaluation metric used is nDCG@10, the official metric of the BEIR benchmark.

Statistical significance analysis was performed using Student's two-tailed paired t-test with corrections, as per common practice in information retrieval. The results of this analysis is reported in Appendix D due to space constraints.

# 4 Results

## 4.1 Main results

We present our main results in Table 1, highlighting key findings. For fair comparison, all the re-rankers consider the top 100 documents retrieved by BM25. Firstly, it is evident that retrievers and re-rankers fine-tuned on MS MARCO training data consistently outperform zero-shot retrievers and QLM

---

[3]These models however employ the self-instruction approach (Wang et al., 2022b), which may involve a small number of randomly generated QG instructions.

[4]Due to limited computational resources and numerous LLMs with various settings to run, and in order to ensure feasibility, we considered a subset of BEIR that includes the most widely used datasets in the literature. Despite being a subset, it comprises a total of 1,347 queries with deep ranking judgments across 4 distinct domains.

re-rankers across all datasets, except for T5-QLM-large, which is based on a smaller T5 model. This outcome is expected since these methods benefit from utilizing extensive human-judged QA training data and the knowledge can be effectively transferred to the datasets we tested.

On the other hand, zero-shot QLMs and QG fine-tuned QLMs exhibit competitive, similar effectiveness. This finding is somewhat surprising, considering that QG fine-tuned QLMs are explicitly trained on QG tasks, making them a form of transfer learning. This finding suggests that pre-trained-only models such as LLaMA and Falcon possess strong zero-shot QLM ranking capabilities.

Another interesting finding is that instruction tuning can hinder LLMs' QLM ranking ability if the QG task is not included in the instruction fine-tuning data. This is evident in the results of Alpaca-7B, StableVicuna-13B, Falcon-7B-instruct and Falcon-40B-instruct, which are instruction-tuned versions of LLaMA-7B, LLaMA-13B, Falcon-7B and Falcon-40B, respectively. Our hypothesis to this unexpected finding is that instruction-tuned models tend to pay more attention to the task instructions and less attention to the input content itself. Although they are good at following instructions in the generation task, the most important information for evaluating query likelihood is in the document content, thus instruction-tuning hurts query likelihood estimation for LLMs. On the other hand, QG instruction-tuned LLMs show large improvements in QLM ranking. For example, the T0 and FlanT5 models are QG-tuned versions of T5 models, and they perform better. These results confirm that T0 and FlanT5 leverage their fine-tuning data, thus should be considered within the transfer learning setting.

In terms of model size, larger LLMs generally tend to be more effective, although there are exceptions. For instance, LLaMA-7B outperforms LLaMA-13B on DBpedia.

## 4.2 Interpolation with BM25

Table 2 demonstrates the impact of interpolating with BM25 scores. Notably, we observe a large decrease in the effectiveness of monoT5 re-rankers, which are trained on large-scale QA domain data, when interpolating with BM25. This finding aligns with a study conducted by Yates et al. (2021). In contrast, QLM re-rankers consistently exhibited higher effectiveness across most datasets when us-

Table 2: Interpolation results. Increased/decreased scores are noted with $\uparrow$ / $\downarrow$.

| Methods | TRECC | DBpedia | FiQA | Robust04 | Avg |
|---|---|---|---|---|---|
| **without interpolation** | | | | | |
| monoT5-3B | 79.8 | 44.8 | 46.0 | 56.2 | 56.7 |
| monoT5-3B-InPars-v2 | 83.8 | 46.6 | 46.1 | 58.5 | 58.8 |
| FlanT5-3B | 72.0 | 37.0 | 41.7 | 47.0 | 49.4 |
| FlanT5-11B | 75.1 | 39.9 | 44.9 | 50.8 | 52.7 |
| LLaMA-7B | 68.0 | 37.5 | 41.8 | 51.6 | 49.7 |
| Falcon-7B | 73.1 | 39.5 | 41.7 | 49.2 | 50.9 |
| **with interpolation** | | | | | |
| monoT5-3B | 66.3$\downarrow$ | 44.6$\downarrow$ | 41.5$\downarrow$ | 55.1$\downarrow$ | 51.9$\downarrow$ |
| monoT5-3B-InPars-v2 | 82.1$\downarrow$ | 45.5$\downarrow$ | 43.5$\downarrow$ | 54.0$\downarrow$ | 56.3$\downarrow$ |
| FlanT5-3B | 71.1$\downarrow$ | 39.7$\uparrow$ | 41.2$\downarrow$ | 50.0$\uparrow$ | 50.5$\uparrow$ |
| FlanT5-11B | 74.9$\downarrow$ | 41.7$\uparrow$ | 43.3$\downarrow$ | 52.4$\uparrow$ | 53.1$\uparrow$ |
| LLaMA-7B | 69.4$\uparrow$ | 39.9$\uparrow$ | 41.5$\downarrow$ | 53.6$\uparrow$ | 51.1$\uparrow$ |
| Falcon-7B | 73.3$\uparrow$ | 41.7$\uparrow$ | 41.3$\downarrow$ | 52.5$\uparrow$ | 52.2$\uparrow$ |

ing interpolation with BM25. It is worth noting that this improvement is (almost) cost-free, as it does not require any additional relevance score estimation; it simply involves linearly interpolating with scores from the first stage.

We note that the results in Table 2 are obtained by setting $\alpha = 0.2$ without tuning this parameter because we are testing our method in zero-shot setting where this parameter needs to be set without validation data. Nonetheless, we conduct a post-hoc analysis on TRECC to understand the sensitivity of this parameter. The results are presented in Figure 1. From the results, we can draw the following conclusions:

1. The interpolation strategy consistently has a negative impact on monoT5-3B, while it consistently benefits instruction-tuned and zero-shot rerankers.

2. Instruction-tuned rerankers consistently underperform their corresponding zero-shot rerankers, regardless of the set alpha value.

3. Optimal values of $\alpha$ for both instruction-tuned and zero-shot rerankers fall within the range of 0.1 to 0.4.

## 4.3 Effective ranking pipeline

In Table 3 we push the boundary of our two-stage QLM ranking pipeline in both zero-shot and few-shot setting to obtain high ranking effectiveness. For this purpose, we use the same linear interpolation as Equation 2 with $\alpha = 0.5$ to combine BM25 and HyDE as the zero-shot first-stage retriever.[5] The top 100 documents retrieved by this hybrid retriever are then re-ranked using QLMs.

---

[5]This value was chosen to provide equal weight to the two components, and no parameter exploration was undertaken.

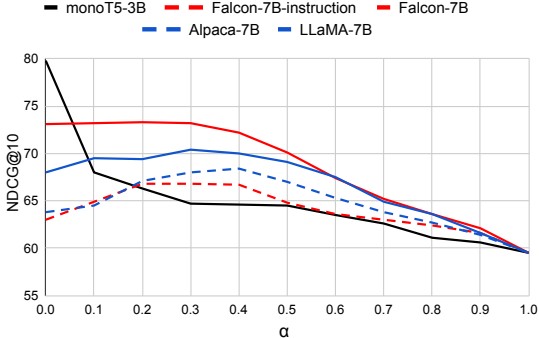

Figure 1: Impact of $\alpha$ on TRECC dataset.

Firstly, our results suggest that the effectiveness of zero-shot first-stage retrieval can be improved by simply interpolating sparse and dense retrievers. Moreover, after QLM re-ranking, the nDCG@10 values surpass those in Table 1. This indicates that zero-shot QLM re-rankers benefit from a stronger first-stage retriever, leading to improved overall ranking effectiveness. For the few-shot results, we observe that providing only three GBQ examples to the model further enhances ranking effectiveness, although this effect is less pronounced for FlanT5. Remarkably, our QLM ranking pipeline achieves nDCG@10 on par with or higher than the state-of-the-art PROMPTAGATOR method on comparable datasets in both zero-shot and few-shot settings. It is important to note that PROMPTAGATOR requires training on a large amount of synthetically generated data for both the retriever and re-ranker, whereas our approach does not require any training. It's worth highlighting that instruction-tuned LLMs continue to exhibit lower effectiveness compared to their pre-trained-only LLMs, even when a better first-stage retriever is employed and under a few-shot setting.

## 5 Conclusion

In this paper, we adapt recent advanced LLMs into QLMs for ranking documents and comprehensively study their zero-shot ranking ability. Our results highlight that these LLMs possess remarkable zero-shot ranking effectiveness. Moreover, we observe that additional instruction fine-tuned LLMs underperformed in this task. This important insight is overlooked in previous studies. Furthermore, our study shows that by integrating LLM-based QLMs with a hybrid zero-shot retriever, a novel state-of-the-art ranking pipeline can be obtained that excels in both

Table 3: Zero-shot/few-shot ranking systems. *PROMPTAGATOR++ re-rankers use their own zero/few-shot PROMPTAGATOR first-stage retrievers, scores are copied from the original paper as the model is not publicly available. Other re-rankers consider the Top100 documents retrieved by BM25 + HyDE.

| Methods | TRECC | DBpedia | FiQA | Robust04 | Avg |
|---|---|---|---|---|---|
| **Zero-shot Retrievers** | | | | | |
| BM25 + HyDE | 69.8 | 41.7 | 30.9 | 49.7 | 48.0 |
| *PROMPTAGATOR | 72.7 | 36.4 | 40.4 | - | - |
| **Few-shot Retrievers** | | | | | |
| *PROMPTAGATOR | 75.6 | 38.0 | 46.2 | - | - |
| **Zero-shot Re-rankers** | | | | | |
| *PROMPTAGATOR++ | 76.0 | 41.3 | 45.9 | - | - |
| FlanT5-11B | 75.8 | 46.2 | 49.5 | 56.6 | 57.0 |
| StableLM-7B | 74.2 | 41.8 | 38.0 | 53.2 | 51.8 |
| Alpaca-7B | 71.6 | 40.1 | 39.5 | 50.9 | 50.5 |
| LLaMA-7B | 72.4 | 45.4 | 46.8 | 57.4 | 55.5 |
| Falcon-7B-instruct | 68.8 | 43.1 | 37.4 | 54.6 | 51.0 |
| Falcon-7B | 76.6 | 46.1 | 45.8 | 55.1 | 55.9 |
| **Few-shot Re-rankers** | | | | | |
| *PROMPTAGATOR++ | 76.2 | 43.4 | 49.4 | - | - |
| FlanT5-11B | 77.2 | 45.1 | 49.7 | 58.2 | 57.6 |
| StableLM-7B | 72.2 | 42.8 | 38.0 | 51.7 | 51.2 |
| Alpaca-7B | 72.3 | 42.5 | 41.8 | 53.0 | 52.4 |
| LLaMA-7B | 77.8 | 47.7 | **50.4** | **59.5** | **58.8** |
| Falcon-7B-instruct | 74.9 | 45.2 | 42.8 | 56.1 | 54.8 |
| Falcon-7B | **78.6** | **48.0** | 48.6 | 59.0 | 58.5 |

zero-shot and few-shot scenarios, showcasing the effectiveness and versatility of LLM-based QLMs.

## Limitations

While theoretically our QLM method can be applied to any LLM, for practical implementation, access to the model output logits is required. Therefore, in this paper, our focus has been solely on open-source LLMs where we can have access to the model weights. In contrast, approaches like Inpars and PROMPTAGATOR, which extract knowledge from the text produced by LLMs, do not require access to the model weights. Common commercial API services that expose popular close-source models such as GPT-4, however, do not provide access to model logits. offered by popular close-source models such as GPT-4 . These can easily be used within Inpars and PROMPTAGATOR by directly leveraging the generated text. However, our method cannot use these models because they do not provide access to the logits. It might be possible that in future commercial LLM provides would add functionalities in their APIs to access model logits.

Our focus on open-source LLMs also offers us the opportunity to scrutinise the data used to train the LLMs to ascertain that no QG data was used. This reassures a genuine zero-shot setting is considered, as opposed to previous work on LLM-based

QLMs (Sachan et al., 2022). Although LLaMA and Falcon are primarily pre-trained using unsupervised learning on unstructured text data, it remains possible that the pre-training data contains text snippets that serve as instructions and labels for the QG task. In order to ascertain the authenticity of the zero-shot setting, it may be necessary to thoroughly analyze and identify such text snippets within the pre-training data.

In the paper, we could not report a complete statistical significance analysis of the results due to space limitation. Appendix D reports a detailed analysis. However, our analysis was limited by the unavailability of run files for some of the models published in previous works, as they were not released by authors. In these cases, we could not perform statistical comparisons with respect to the runs we produced. We note this is a common problem when authors do not release their models' runs. We make all run files available, along with code, at https://github.com/ielab/llm-qlm.

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

Table 4: Prompts used for each LLM-dataset pair. For Alpaca-7B and StableLM-7B we also prepend a system prompt according to the fine-tuning recipe of the each model. For Alpaca-7B is "Below is an instruction that describes a task, paired with an input that provides further context. Write a response that appropriately completes the request.\n\n". For SableLM-7B is "<|SYSTEM|># StableLM Tuned (Alpha version)\n- StableLM is a helpful and harmless open-source AI language model developed by StabilityAI.\n- StableLM is excited to be able to help the user, but will refuse to do anything that could be considered harmful to the user.\n- StableLM is more than just an information source, StableLM is also able to write poetry, short stories, and make jokes.\n- StableLM will refuse to participate in anything that could harm a human.\n"

| LLMs | TRECC | DBpedia | FiQA | Robust04 |
|---|---|---|---|---|
| T5-3B/T5-11B/FlanT5-3B/FlanT5-11B | Generate a question that is the most relevant to the given article's title and abstract.\n{doc} | Generate a query that includes an entity and is also highly relevant to the given Wikipedia page title and abstract.\n{doc} | Generate a question that is the most relevant to the given document.\n{doc} | Generate a question that is the most relevant to the given document.\n{doc} |
| T0-3B/T0-11B | Please write a question based on this passage.\n{doc} | Please write a question based on this passage.\n{doc} | Please write a question based on this passage.\n{doc} | Please write a question based on this passage.\n{doc} |
| LLaMA-7B/LLaMA13B/Falcon-7B/Falcon-13B/Falcon-7B-instruct/Falcon-13B-instruct | Generate a question that is the most relevant to the given article's title and abstract.\n{doc}\n\nHere is a generated relevant question: | Generate a query that includes an entity and is also highly relevant to the given Wikipedia page title and abstract.\n{doc}\n\nHere is a generated relevant question: | Generate a question that is the most relevant to the given document.\nThe document: {doc}\n\nHere is a generated relevant question: | Generate a question that is the most relevant to the given document.\nThe document: {doc}\n\nHere is a generated relevant question: |
| Alpaca-7B | ### Instruction:\nGenerate a question that is the most relevant to the given article's title and abstract.\n\n### Input:\n{doc}\n\n### Response: | ### Instruction:\nGenerate a query that includes an entity and is also highly relevant to the given Wikipedia page title and abstract.\n\n### Input:\n{doc}\n\n###Response: | ### Instruction:\nGenerate a question that is the most relevant to the given document.\n\n### Input:\n{doc}\n\n### Response: | ### Instruction:\nGenerate a question that is the most relevant to the given document.\n\n### Input:\n{doc}\n\n### Response: |
| StableLM-7B | <|USER|>Generate a question that is the most relevant to the given article's title and abstract.\n{doc}<|ASSISTANT|>Here is a generated relevant question: | <|USER|>Generate a query that includes an entity and is also highly relevant to the given Wikipedia page title and abstract.\n{doc}<|ASSISTANT|>Here is a generated relevant question: | <|USER|>Generate a question that is the most relevant to the given document.\n{doc}<|ASSISTANT|>Here is a generated relevant question: | <|USER|>Generate a question that is the most relevant to the given document.\nThe document: {doc}<|ASSISTANT|>Here is a generated relevant question |
| StableVicuna-13B | ### Human: Generate a question that is the most relevant to the given article's title and abstract.\n{doc}\n### Assistant: Here is a generated relevant question: | ### Human: Generate a query that includes an entity and is also highly relevant to the given Wikipedia page title and abstract.\n{doc}\n### Assistant: Here is a generated relevant question: | ### Human: Generate a question that is the most relevant to the given document.\nThe document: {doc}\n### Assistant: Here is a generated relevant question: | ### Human: Generate a question that is the most relevant to the given document.\nThe document: {doc}\n### Assistant: Here is a generated relevant question: |

## A    Models and datasets prompts

Given that various instruction-tuned LLMs might be fine-tuned using diverse system and instruction prompts, coupled with the fact that datasets vary in document formats across different domains, it becomes necessary to employ specific prompts tailored to each LLM-dataset pair to achieve optimal zero-shot ranking performance. Thus, we design a prompt for each LLM-dataset pair based on the LLM usage instruction provided by the original authors and dataset features. To facilitate clarity, we have compiled a comprehensive list of all the prompts utilized for each LLM-dataset pair, which can be found in Table 4.

## B    List of Huggingface model names

Table 5 provides links to the Huggingface model hub (Wolf et al., 2020) for the LLMs used in this paper. All the models can be conveniently downloaded directly from the Huggingface model hub, with the exception of Alpaca-7B. For Alpaca-7B, we followed an open-sourced github repository to perform the fine-tuning of LLaMA-7B ourselves.

## C    Descriptions of Baselines

- **BM25** (Robertson and Zaragoza, 2009): A widely used statistical bag-of-words approach

Table 5: Huggingface model hub links for LLMs used in this paper.

| LLMs | Link |
|---|---|
| T5-3B | t5-3b |
| T5-11B | t5-11b |
| StableLM-7B | stabilityai/stablelm-tuned-alpha-7b |
| StableVicuna-13B | TheBloke/stable-vicuna-13B-HF |
| Falcon-7B | tiiuae/falcon-7b |
| Falcon-7B-instruct | tiiuae/falcon-7b-instruct |
| Falcon-40B | tiiuae/falcon-40b |
| Falcon-40B-instruct | tiiuae/falcon-40b-instruct |
| T0-3B | bigscience/T0_3B |
| T0-11B | bigscience/T0 |
| FlanT5-3B | google/flan-t5-xl |
| FlanT5-11B | google/flan-t5-xxl |
| LLaMA-7B | huggyllama/llama-7b |
| LLaMA-13B | huggyllama/llama-13b |
| Alpaca-7B | https://github.com/tatsu-lab/stanford_alpaca |

Table 6: Overall effectiveness of the models and statistical significance analysis. The best results are highlighted in boldface. Superscripts denote significant differences (t-test, $p \leq 0.05$). $x \rightarrow y$ denotes the $x$ retriever re-ranked by $y$ re-ranker.

| # | Model | TRECC | DBpedia | FiQA | Robust04 |
|---|---|---|---|---|---|
| a | BM25 | $59.5^b$ | $31.8^b$ | $23.6^b$ | $40.7^e$ |
| b | QLM-Dirichlet | $50.8$ | $29.5$ | $20.5$ | $40.7^e$ |
| c | HyDE | $58.2$ | $37.1^{abe}$ | $26.6^{ab}$ | $41.8^e$ |
| d | BM25+HyDE | $69.9^{abc}$ | $41.6^{abcefghiklmop}$ | $30.9^{abc}$ | $49.7^{abcef}$ |
| e | BM25 -> T5-11B | $67.9^{abc}$ | $33.7^{ab}$ | $32.0^{abc}$ | $27.4$ |
| f | BM25 -> Alpaca-7B | $67.0^{abc}$ | $35.0^{ab}$ | $33.7^{abcd}$ | $44.6^{abe}$ |
| g | BM25 -> StableLM-7B | $74.0^{abcefi}$ | $37.2^{abef}$ | $34.1^{abcde}$ | $48.3^{abcef}$ |
| h | BM25 -> StableVicuna-13B | $71.8^{abcfi}$ | $39.4^{abefgp}$ | $39.1^{abcdefgi}$ | $51.3^{abcefg}$ |
| i | BM25 -> Falcon-7B-instruct | $66.8^{abc}$ | $38.2^{abef}$ | $33.4^{abcd}$ | $50.7^{abcefg}$ |
| j | BM25 -> Falcon-40B-instruct | $70.2^{abc}$ | $40.5^{abcefgiklp}$ | $40.8^{abcdefghi}$ | $51.3^{abcefg}$ |
| k | BM25 -> T0-3B | $71.6^{abc}$ | $38.8^{abefg}$ | $41.4^{abcdefghi}$ | $50.1^{abcefg}$ |
| l | BM25 -> T0-11B | $73.9^{abcefijop}$ | $38.7^{abefg}$ | $43.8^{abcdefghijkmopq}$ | $49.7^{abcef}$ |
| m | BM25 -> FlanT5-3B | $71.1^{abc}$ | $39.7^{abcefgip}$ | $41.2^{abcdefghi}$ | $50.0^{abcefg}$ |
| n | BM25 -> FlanT5-11B | $74.9^{abcdefijkmop}$ | $41.7^{abcefghiklmop}$ | $43.3^{abcdefghijkmopq}$ | $52.4^{abcdefgiklm}$ |
| o | BM25 -> LLaMA-7B | $69.4^{abc}$ | $39.9^{abcefgip}$ | $41.5^{abcdefghi}$ | $53.6^{abcdefghijklm}$ |
| p | BM25 -> LLaMA-13B | $69.8^{abc}$ | $37.6^{abef}$ | $41.8^{abcdefghi}$ | $54.2^{abcdefghijklmn}$ |
| q | BM25 -> Falcon-7B | $73.3^{abcefiop}$ | $41.4^{abcefghiklmop}$ | $41.2^{abcdefghi}$ | $52.5^{abcdefgiklm}$ |
| r | BM25 -> Falcon-40B | $75.2^{abcefijkmop}$ | $41.0^{abcefghiklop}$ | $43.1^{abcdefghijkmopq}$ | $53.1^{abcdefghijklm}$ |
| s | BM25 -> monoT5-3B | $79.8^{abcdefghijklmopqv}$ | $44.8^{abcdefghijklmnopqr}$ | $46.0^{abcdefghijklmnopqr}$ | $56.2^{abcdefghijklmnopqr}$ |
| t | BM25 -> monoT5-3B-InPars-v2 | $83.7^{abcdefghijklmnopqrsuvwxyz}$ | $46.5^{abcdefghijklmnopqrs}$ | $46.1^{abcdefghijklmnopqr}$ | $58.5^{abcdefghijklmnopqrsw}$ |
| u | BM25+HyDE -> FlanT5-11B | $75.8^{abcdefijkmop}$ | $46.2^{abcdefghijklmnopqrx}$ | $49.5^{abcdefghijklmnopqrstvw}$ | $56.6^{abcdefghijklmnoqr}$ |
| v | BM25+HyDE -> LLaMA-7B | $72.4^{abc}$ | $45.4^{abcdefghijklmnopqr}$ | $46.8^{abcdefghijklmnopqr}$ | $57.4^{abcdefghijklmnopqrw}$ |
| w | BM25+HyDE -> Falcon-7B | $76.6^{abcdefijkmopv}$ | $46.1^{abcdefghijklmnopqr}$ | $45.8^{abcdefghijklmnopqr}$ | $55.1^{abcdefghijklmq}$ |
| x | BM25+HyDE -> FlanT5-11B-fewshot | $77.2^{abcdefhijklmopuv}$ | $45.1^{abcdefghijklmnopqr}$ | $49.7^{abcdefghijklmnopqrstvw}$ | $58.3^{abcdefghijklmnopqruw}$ |
| y | BM25+HyDE -> LLaMA-7B-fewshot | $77.8^{abcdefhijklmopv}$ | $47.7^{abcdefghijklmnopqrsuvwx}$ | $\mathbf{50.4}^{abcdefghijklmnopqrstvwz}$ | $\mathbf{59.5}^{abcdefghijklmnopqrsuvw}$ |
| z | BM25+HyDE -> Falcon-7B-fewshot | $78.6^{abcdefghijklmopqv}$ | $\mathbf{48.0}^{abcdefghijklmnopqrsuvwx}$ | $48.6^{abcdefghijklmnopqrstvw}$ | $59.0^{abcdefghijklmnopqrsuvw}$ |

that is commonly used as the zero-shot first-stage retrieval method. We use the Pyserini "two-click reproductions" (Ma et al., 2022) to produce the BM25 results on BEIR datasets.

- **QLM-Dirichlet** (Zhai and Lafferty, 2001): The traditional QLM method that exploits term statistics and Dirichlet smoothing technique to estimate query likelihood, we also use Pyserini implementation for this baseline.

- **Contriever** (Izacard et al., 2022): A zero-shot dense retriever that pre-trained on text paragraphs with unsupervised contrastive learning.

- **HyDE** (Gao et al., 2022): A two-step zero-shot first-stage retriever that leverages generative LLMs and Contriever. In the first step, a prompt is provided to a LLM to generate multiple documents relevant to the given query. Subsequently, in the second step, the generated documents are encoded into vectors using the Contriever query encoder and then aggregated to form a new query vector for the search process. We utilized the open-sourced implementation provided by the original authors for our experiments: https://github.com/texttron/hyde.

- **Contriver-msmarco**. A Contriever checkpoint further pre-trained on MS MARCO training data. We use the Pyserini provided

pre-build dense vector index and model checkpoint for this baseline.

- **SPLADE-distill** (Formal et al., 2022): A first-stage sparse retrieval model that exploits BERT PLM to learn query/document sparse term expansion and weights. We use the Pyserini provided pre-build index and SPLADE checkpoint to produce the results.

- **DRAGON+** (Lin et al., 2023): A dense retriever model that fine-tuned on augmented MS MARCO corpus and uses multiple retrievers to conduct automatical relevance labeling. It stands as the current state-of-the-art dense retriever in the transfer learning setting. We use the scores reported on the BEIR learderboard [6] for this baseline.

- **T5-QLM-large** (Zhuang et al., 2021): A T5-based QLM method that fine-tuned on MS MARCO QG training data. We use the implement this method with open-sourced docTquery-T5 (Nogueira and Lin, 2019) checkpoint [7].

- **monoT5-3B** (Nogueira et al., 2020). A T5-based cross-encoder re-ranker that fine-tuned on MS MARCO training data. We use the

---

[6]https://eval.ai/web/challenges/challenge-page/1897/leaderboard/4475
[7]castorini/doc2query-t5-large-msmarco

open-sourced implementation provided by In-pars authors [8].

- **monoT5-3B-Inpars-v2** (Jeronymo et al., 2023): A T5-based cross-encoder re-ranker that fine-tuned on MS MARCO training data and in-domain synthetic queries that generated by LLMs. It is the current state-of-the-art re-ranker in transfer learning setting. We use the open-sourced implementation provided by the original authors [9].

- **PROMPTAGATOR**(Dai et al., 2023): These methods consist of a Transformer encoder-based retriever and re-ranker that are trained using synthetic queries generated by LLMs. They offer both zero-shot and few-shot settings. As public model checkpoints are not currently available, we refer to the scores reported in the original paper as our point of reference for comparing against our own methods and baselines.

## D Statistical significance analysis

In Table 6 we report a statistical significance analysis for all the methods for which we can obtain a run file, along with our methods. The analysis was performed using the Student's two-tailed paired t-test with corrections, as per common practice in information retrieval. We used the Python toolkit ranx (Bassani and Romelli, 2022) for generating the report.

---

[8] https://github.com/zetaalphavector/InPars
[9] https://github.com/zetaalphavector/InPars