# OpenReview forum: "Open-source Large Language Models are Strong Zero-shot Query Likelihood Models for Document Ranking"
_EMNLP/2023/Conference — EMNLP 2023 Findings_

### Official Review · Reviewer_ia7W · 2023-08-04

**Typos Grammar Style And Presentation Improvements:** N/A
**Soundness:** 4

**Excitement:**

3: Ambivalent: It has merits (e.g., it reports state-of-the-art results, the idea is nice), but there are key weaknesses (e.g., it describes incremental work), and it can significantly benefit from another round of revision. However, I won't object to accepting it if my co-reviewers champion it.

**Missing References:**

N/A

**Paper Topic And Main Contributions:**

This paper investigates the zero-shot ranking effectiveness of large language models (LLMs) in information retrieval. It finds that LLMs have robust zero-shot ranking abilities without fine-tuning. The paper proposes a ranking system that combines LLM-based query language models (QLMs) with a hybrid zero-shot retriever, achieving state-of-the-art results. Limitations include the requirement for open-source LLMs and the need for further scrutiny of training data for genuine zero-shot settings.

**Questions For The Authors:**

Address the concerns in "Reasons To Reject"

**Reasons To Accept:**

1. The paper provides a comprehensive study on the zero-shot ranking ability of advanced language models (LLMs) converted into query language models (QLMs). This study highlights the remarkable zero-shot ranking effectiveness of LLM-based QLMs without the need for additional instruction fine-tuning. This insight is important and overlooked in previous studies.

2. The paper demonstrates that by integrating LLM-based QLMs with a hybrid zero-shot retriever, a novel state-of-the-art ranking pipeline can be obtained that excels in both zero-shot and few-shot scenarios. This showcases the effectiveness and versatility of LLM-based QLMs.

3. The paper presents key findings that reveal the competitive effectiveness of zero-shot QLMs and QG fine-tuned QLMs. This finding is surprising considering that QG fine-tuned QLMs are explicitly trained on QG tasks, making them a form of transfer learning. The paper suggests that pre-trained-only models such as LLaMA and Falcon possess strong zero-shot QLM ranking capabilities.

Overall, the paper contributes valuable insights into the zero-shot ranking ability of LLM-based QLMs.

**Reasons To Reject:**

1. One weakness of the paper is the limited focus on open-source LLMs, which restricts the applicability of the proposed QLM method. The paper acknowledges that access to the model output logits is required for practical implementation, which limits the method to open-source LLMs. This limitation overlooks the potential of utilizing powerful close-source models, such as GPT-4, which could meet the requirements of other approaches like Inpars and PROMPTAGATOR.

2. Another weakness is the lack of a complete statistical significance analysis of the results due to space limitations. While the paper includes a detailed analysis in the appendix, the unavailability of run files for some models from previous works prevents statistical comparisons with the runs produced in this study. This limitation hinders a comprehensive evaluation and comparison of the proposed approach with previous methods.

3. The paper does not thoroughly address the authenticity of the zero-shot setting in the pre-training data of LLaMA and Falcon. Although the paper reassures that no QG data was used in the pre-training of these models, it suggests the need for a thorough analysis to identify any text snippets that may serve as instructions and labels for the QG.

**Reproducibility:**

5: Could easily reproduce the results.

**Reviewer Confidence:**

3: Pretty sure, but there's a chance I missed something. Although I have a good feel for this area in general, I did not carefully check the paper's details, e.g., the math, experimental design, or novelty.

---

> ### Author Rebuttal · Authors · 2023-08-29
>
> Thanks for acknowledging our surprising findings and think our experiments provide sufficient support for all of its claims/arguments.
> ***
> ### Response to reject reason 1:
> > "One weakness of the paper is the limited focus on open-source LLMs, which restricts the applicability of the proposed QLM method. The paper acknowledges that access to the model output logits is required for practical implementation, which limits the method to open-source LLMs. This limitation overlooks the potential of utilizing powerful close-source models, such as GPT-4, which could meet the requirements of other approaches like Inpars and PROMPTAGATOR."
>
> In fact, theoretically, our method can be applied to any LLMs. The only reason our work is currently limited to open-sourced LLMs is that the required API is not available yet. This limitation is beyond our control. We hope that our work will encourage the NLP community to embrace open-sourcing such models. As demonstrated by our work, open-sourcing can lead to many more opportunities for research.
> ***
> ### Response to reject reason 2:
> > "Another weakness is the lack of a complete statistical significance analysis of the results due to space limitations. While the paper includes a detailed analysis in the appendix, the unavailability of run files for some models from previous works prevents statistical comparisons with the runs produced in this study. This limitation hinders a comprehensive evaluation and comparison of the proposed approach with previous methods."
>
> We tried our best to acquire baseline run files to perform statistical significance tests against our method. Regrettably, we encountered difficulties in obtaining run files for a few baselines. This was due to either unavailability of the model checkpoint or the code implementation not being publicly accessible at this time. Once again, we hope that the community will increasingly embrace open-sourcing by sharing both models and code. Upon acceptance, we will promptly make our code and comprehensive instructions available to the public, enabling full reproducibility of our results.
> ***
> ### Response to reject reason 3:
> > "The paper does not thoroughly address the authenticity of the zero-shot setting in the pre-training data of LLaMA and Falcon. Although the paper reassures that no QG data was used in the pre-training of these models, it suggests the need for a thorough analysis to identify any text snippets that may serve as instructions and labels for the QG."
>
> The LLaMA and Falcon models are pre-trained on unstructured, free text. It is improbable that the pre-training data will contain numerous text snippets structured as question-generation instruction data by random chance. Although this possibility does exist, the portion of this kind of data is definitely extremely small. On the other hand, T0 and Flan-T5 have a subset of fine-tuning data that specifically instructs the model to do the question generation. Thus, as we argued in the paper, they cannot be considered as zero-shot QLM rankers.

---

### Official Review · Reviewer_Zpce · 2023-08-11

**Soundness:** 2

**Excitement:**

3: Ambivalent: It has merits (e.g., it reports state-of-the-art results, the idea is nice), but there are key weaknesses (e.g., it describes incremental work), and it can significantly benefit from another round of revision. However, I won't object to accepting it if my co-reviewers champion it.

**Paper Topic And Main Contributions:**

This work studies the zero-shot ranking capability of LLMs pre-trained on unsupervised text. Specifically, this work looks at LLMs that are query likelihood models (QLMs): estimate the probability that the given document can answer the given query, and those probability scores are used to produce a ranking over candidate documents given the query. The motivation is to study QLMs’ zero-shot ability with prompting, addressing that some LLMs are not truly zero-shot due to the possible presence of question generation datasets in the pre-training.

Experiments are to compare with a number of instruction-tuned LLM-based retrievers and rerankers, as well as some few-shot models, on 4 BEIR datasets. I found the conclusions from the experimental results are confusing, since the trend looks very mixed. It also looks very strange that interpolating with first-stage retrievers worsens the performance, and I am very curious about why. Table 3 compares zero-shot and few-shot setup with a better first-stage retrieval, but the few-shot rerankers seem to work better than zero-shot, which makes the claim on “comparable performance” a bit misleading.

Overall, the work is well-motivated in terms of studying zero-shot performance with models that are not pre-trained on related tasks, but the experiments are less supportive of their claims in terms of general claims and performance trends.


**Questions For The Authors:**

Why not include other publicly available BEIR benchmarks in the evaluation?

**Reasons To Accept:**

Please see the main review.

**Reasons To Reject:**

Please see the main review.

**Reproducibility:**

5: Could easily reproduce the results.

**Reviewer Confidence:**

4: Quite sure. I tried to check the important points carefully. It's unlikely, though conceivable, that I missed something that should affect my ratings.

---

> ### Author Rebuttal · Authors · 2023-08-29
>
> Thank you for acknowledging that our work is well-motivated.
> ***
> ### Response to reviewer remark1:
> > "I found the conclusions from the experimental results are confusing, since the trend looks very mixed."
>
> We believe our results are considerably consistent. Our main claim, that instruction-tuned LLMs are inferior to their pretrained-only base LLMs, is supported by the results of the following LLM pairs: (LLaMA-7B, Alpaca-7B), (LLaMA-13B, StableVicuna-13B), (Falcon-7B, Falcon-7B-instruct), and (Falcon-40B, Falcon-40B-instruct). In total, we have 16 pairs across 4 datasets; only 2 out of 16 do not support the claim (specifically, the LLaMA-13B, and StableVicuna-13B pair on TRECC and DBpedia). Moreover, the best zero-shot and few-shot performance is achieved by pretrained-only LLMs. Additionally, the additional results in our response to reviewer #2 further support our claims. Thus, we believe our results are sufficient to uphold our claims
> ***
> ### Response to reviewer remark2:
> > "It also looks very strange that interpolating with first-stage retrievers worsens the performance, and I am very curious about why."
>
> We assume this refers to our results of fine-tuned models (monoT5-3B and monoT5-3B-InPars-v2) in table2, (also our additional results in reviewer#2 remark2’s response hold the same trend). In fact, our results also align with a previous study (https://arxiv.org/pdf/2010.06467.pdf , figure 10). We believe this is due to the fact that these rerankers are fine-tuned on hard negatives sampled from BM25, thus they have been exposed to documents retrieved from the first-stage retriever during training and have learned how to rank BM25 ranking. Hence, the BM25 scores have limited help in this situation.
>
> ***
> ### Response to reviewer remark3:
> > "Table 3 compares zero-shot and few-shot setup with a better first-stage retrieval, but the few-shot rerankers seem to work better than zero-shot, which makes the claim on “comparable performance” a bit misleading."
>
> We apologize for the confusion. In our paper, we did not intend to convey that few-shot rerankers have comparable performance to zero-shot rerankers. Rather, we meant to highlight that our tested LLMs exhibit comparable or higher performance to PROMPTAGATOR on similar datasets, both in zero-shot and few-shot settings. We will ensure that our points are clarified in the revised version.

---

### Official Review · Reviewer_vsPL · 2023-08-12

**Soundness:** 2

**Excitement:**

3: Ambivalent: It has merits (e.g., it reports state-of-the-art results, the idea is nice), but there are key weaknesses (e.g., it describes incremental work), and it can significantly benefit from another round of revision. However, I won't object to accepting it if my co-reviewers champion it.

**Missing References:**

1. SimLM (https://arxiv.org/abs/2207.02578) can be added as a zero-shot first-stage retrieval method.

**Paper Topic And Main Contributions:**

This paper investigates the effectiveness of using pre-trained LLMs that are not instruction fine-tuned as Query Likelihood Models (QLMs) for zero-shot document ranking task. They empirically show that the additional instruction fine-tuning of LLMs degrades their ranking performance unless question generation task is present in fine-tuning dataset. They follow the settings from [1,2] to use LLMs as zero-shot QLM re-ranker and interpolate the first-stage retriever scores with QLM scores inspired by [3]. They also show that using the hybrid zero-shot first-stage retrieval system improves the performance further. They explore the few-shot QLM re-ranker setting following the prompt template introduced in [4].

References:
1. https://aclanthology.org/2022.emnlp-main.249/
2. https://link.springer.com/chapter/10.1007/978-3-030-72240-1_49
3. https://dl.acm.org/doi/abs/10.1145/3471158.3472233
4. https://arxiv.org/pdf/2202.05144.pdf

**Reasons To Accept:**

A key finding of this paper is that the the pre-trained LLMs are strong zero-shot re-rankers without needing additional fine-tuning. They find that additional instruction fine-tuning degrades the ranking performance if question-generation task is not present in fine-tuning data. This insight can be useful in understanding the workings of LLMs. It also shows that LLMs are strong QLMs for zero-shot ranking. The paper benchmarks a wide range of open-source LLMs in its experiments.

**Reasons To Reject:**

1. Paper claims that [1] does not meet the "genuine zero-shot" scenario a T0 has undergone fine-tuning on question generation (QG) tasks and there is a task leakage to downstream QLM ranking task. Lines 078 to 087. This is one of the key arguments of this paper, but it is not explained/proved in the paper why undergoing fine-tuning on question generation (QG) tasks disqualifies it as "genuine" zero-shot approach, nor it is supported by any references
2. The weights for interpolation strategy is set to 0.2 without conducting any grid search (Line 158). No explanation is provided for the choice of the value for this parameter. The main results for this paper's claims depend on the choice of this parameter. Thus, it is important to find the optimal value of this parameter for different LLMs in different settings. It is possible that the key finding of the paper that the instruction fine-tuning degrades the ranking performance maybe because 0.2 value may be better for non-instruction-tuned LLM and worse for the instruction-tuned LLM, and choosing optimal value of this parameter may carry a fair comparison.
3. Detailed discussion is lacking in the paper on its empirical observation to analyze why the further instruction fine-tuning degrades the ranking performance if question-generation task is not present in fine-tuning data. It would be interesting to understand the theoretical basis behind such behavior.
4. Lacks original ideas in the contributions: (i) Main QLM setting is the same as introduced in [1,2]. (ii) The interpolation of relevancy scores with first-stage retriever is taken as [3]. (iii) The prompt-template for few-shot template is taken from [4]. The paper carries out the ranking experiments using some newer LLMs which were not explored in [1,2].
5. It would be important to have the experimental results using different types of first-stage retrievers other than BM25 too, in order to validate if the observation that instruction-tuning hinders performance if QG task is not included, is consistent in different settings.
6. The experiments are performed only on a small subset (4 datasets) of BEIR benchmark. It is not sufficient to support the claims made in this paper. Thorough evaluation on a larger set is important to support the paper's claims.

References:
1. https://aclanthology.org/2022.emnlp-main.249/
2. https://link.springer.com/chapter/10.1007/978-3-030-72240-1_49
3. https://dl.acm.org/doi/abs/10.1145/3471158.3472233
4. https://arxiv.org/pdf/2202.05144.pdf

**Reproducibility:**

4: Could mostly reproduce the results, but there may be some variation because of sample variance or minor variations in their interpretation of the protocol or method.

**Reviewer Confidence:**

4: Quite sure. I tried to check the important points carefully. It's unlikely, though conceivable, that I missed something that should affect my ratings.

---

> ### Author Rebuttal · Authors · 2023-08-29
>
> Thank you for recognizing that the insights presented in our paper hold value for the community.
> ***
> ### Response to reject reason 1:
> > "Paper claims that [1] does not meet the "genuine zero-shot" scenario a T0 has undergone fine-tuning on question generation (QG) tasks and there is a task leakage to downstream QLM ranking task. Lines 078 to 087. This is one of the key arguments of this paper, but it is not explained/proved in the paper why undergoing fine-tuning on question generation (QG) tasks disqualifies it as "genuine" zero-shot approach, nor it is supported by any references"
>
> We recognise that in the IR literature, there is actually not a clear and well-agreed-upon definition of ‘zero shot’.
>
> The model of [1] is a Query Likelihood ranking approach: estimate the probability of a query given a document. The question generation task is close to estimating the likelihood of a query given a document: given a document, generate a question for this document. This generation is done by maximising the likelihood of the tokens in the relevant question. Thus we see the question generation task as one of optimising query likelihood. Therefore, the two tasks are computationally very similar to each other, and a model fine-tuned for one task will be adept at the other. We believe this situation is more appropriately regarded as transfer learning rather than zero shot (T0 is finetuned on other QA datasets and tested on BEIR).
>
> To address this comment, we will add a section where we define the differences between these settings and we highlight our motivations for not considering the method of [1] as a true zero shot approach.
> ***
> ### Response to reject reason 2:
> > "The weights for interpolation strategy is set to 0.2 without conducting any grid search (Line 158). No explanation is provided for the choice of the value for this parameter. The main results for this paper's claims depend on the choice of this parameter. Thus, it is important to find the optimal value of this parameter for different LLMs in different settings. It is possible that the key finding of the paper that the instruction fine-tuning degrades the ranking performance maybe because 0.2 value may be better for non-instruction-tuned LLM and worse for the instruction-tuned LLM, and choosing optimal value of this parameter may carry a fair comparison."
>
> Our paper sets out to test the real-life scenario of someone deploying LLMs in a real zero shot setting, where this parameter needs to be set without validation data (thus, without tuning the parameter itself). We set the parameter to 0.2 as a reasonable value: give more weight to the second stage re-ranker because this is a better model. We will add this reasoning to the paper.
>
> I do understand the reviewer's concern regarding the sensitivity of this parameter so we did conduct a linear search of alpha values for fine-tuned rerankers (monoT5-3B), instruction-tuned rerankers (Falcon-7B-instruct, Alpaca-7B), and Zero-shot rerankers (Falcon-7B, LLaMA-7B) on TRECC. The results are presented in the table below:
>
> | Methods\Alpha | 0.0 | 0.1 | 0.2 | 0.3 | 0.4 | 0.5 | 0.6 | 0.7 | 0.8 | 0.9 | 1.0 |
> |-------------------|-----|-----|-----|-----|-----|-----|-----|-----|-----|-----|-----|
> | monoT5-3B | 79.8 | 68.0 | 66.3 | 64.7 | 64.6 | 64.5 | 63.5 | 62.6 | 61.1 | 60.6 | 59.5 |
> | Falcon-7B-instruct | 63.0 | 64.9 | 66.8 | 66.8 | 66.7 | 64.8 | 63.6 | 63.0 | 62.4 | 61.6 | 59.5 |
> | Falcon-7B | 73.1 | 73.2 | 73.3 | 73.2 | 72.2 | 70.1 | 67.4 | 65.2 | 63.6 | 62.1 | 59.5 |
> | Alpaca-7B | 63.8 | 64.5 | 67.1 | 68.0 | 68.4 | 67.0 | 65.3 | 63.8 | 62.7 | 61.4 | 59.5 |
> | LLaMA-7B | 68.0 | 69.5 | 69.4 | 70.4 | 70.0 | 69.1 | 67.5 | 64.9 | 63.6 | 61.6 | 59.5 |
>
>
> From the table of results, we can draw the following conclusions:
> 1. The interpolation strategy consistently has a negative impact on monoT5-3B, while it consistently benefits instruction-tuned and zero-shot rerankers.
> 2. Instruction-tuned rerankers consistently underperform their corresponding zero-shot rerankers, regardless of the set alpha value.
> 3. Optimal alpha values for both instruction-tuned and zero-shot rerankers fall within the range of 0.1 to 0.4.
>
> These results provide further support for the claims presented in our paper. We will also include this post-hoc analysis of the parameter's effect in the appendix.
> ***
> ### Response to reject reason 3:
> > "Detailed discussion is lacking in the paper on its empirical observation to analyze why the further instruction fine-tuning degrades the ranking performance if question-generation task is not present in fine-tuning data. It would be interesting to understand the theoretical basis behind such behavior."
>
> We agree this is an unexpected and intriguing result. In fact, we think we have touched on an unexplored and very interesting behavior of LLMs.
>
> We have tried many generation examples with Alpaca and LLaMA and find that the instruction-tuned LLMs (Aplaca) are very good at following instructions: they will generate queries with good quality as expected; however, as our experiments shown, Instruction-tuning seems will hurt the token likelihood distribution of LLMs – it downgrades the quality of next token likelihood estimation. On the other hand, pretrained-only LLMs (LLaMA) are not good at following instructions – if no in-context examples are given, it tends to generate unexpected text, for example, it tends to just complete the prompt or generate unexpected format of content. However, our experiments show that they are better token likelihoods estimators.
>
> We do have a hypothesis of why further instruction-tuning hurts the query likelihood estimation: Instruction-tuned models tend to pay more attention to the task instructions and less attention to the input content itself. Although they are good at following instructions in the generation task, the most important information for evaluating query likelihood is in the doc content, thus instruction-tuning hurts query likelihood estimation for LLMs. We will add this rationale to the paper.
>
> We agree that understanding why this result is so, is a very important topic. We were unable to determine ourselves how this investigation could be answered, and thus wanted to share this result with the community. The aim of this paper was to highlight this issue so that it can be further investigated. We hope our paper will inspire further works on this.
> ***
> ### Response to reject reason 4:
> > "Lacks original ideas in the contributions: (i) Main QLM setting is the same as introduced in [1,2]. (ii) The interpolation of relevancy scores with first-stage retriever is taken as [3]. (iii) The prompt-template for few-shot template is taken from [4]. The paper carries out the ranking experiments using some newer LLMs which were not explored in [1,2]."
>
> We acknowledge that our work builds upon several previous methods. However, our novel contributions can be summarized in the following three aspects:
> 1. Identification and definition of "flaws" in the previous method (not genuine zero-shot).
> 2. Discovery of an intriguing finding: instruction-tuned LLMs perform worse than pretrained-only LLMs in QLM ranking tasks.
> 3. Demonstrate that by employing a simple interpolation method with zero-shot QLM LLM rerankers, we can achieve SOTA zero-shot ranking effectiveness.
> ***
> ### Response to reject reason 5:
> > "It would be important to have the experimental results using different types of first-stage retrievers other than BM25 too, in order to validate if the observation that instruction-tuning hinders performance if QG task is not included, is consistent in different settings."
>
> We conducted additional experiments using instruction-tuned LLMs with BM25 + HyDE, a sparse-dense hybrid that serves as a more effective first-stage retriever. The results, including those without instruction-tuned LLMs, are presented below:
>
> Zeroshot:
> | Model                  | trec-covid | DBpedia | FiQA  | Robust04 | Avg   |
> |------------------------|------------|---------|-------|----------|-------|
> | StableLM-7B            | 74.2       | 41.8    | 38.0  | 53.2     | 51.8  |
> | Alpaca-7B              | 71.6       | 40.1    | 39.5  | 50.9     | 50.5  |
> | LLaMA-7B               | 72.4       | 45.4    | 46.8  | 57.4     | 55.5  |
> | Falcon-7B-insturciton  | 68.8       | 43.1    | 37.4  | 54.6     | 51.0  |
> | Falcon-7B              | 76.6       | 46.1    | 45.8  | 55.1     | 55.9  |
>
> few-shot:
> | Model                  | trec-covid | DBpedia | FiQA  | Robust04 | Avg   |
> |------------------------|------------|---------|-------|----------|-------|
> | StableLM-7B | 72.2       | 42.8    | 38.0  | 51.7     | 51.2  |
> | Alpaca-7B | 72.3       | 42.5    | 41.8  | 53.0     | 52.4  |
> | LLaMA-7B    | 77.8       | 47.7    | 50.4  | 59.5     | 58.8  |
> | Falcon-7B-insturciton  | 74.9       | 45.2    | 42.8  | 56.1     | 54.8  |
> | Falcon-7B   | 78.6       | 48.0    | 48.6  | 59.0     | 58.5  |
>
> The conclusion remains unchanged: instruction-tuned LLMs perform worse than their non-instruction-tuned counterparts in both zero-shot and few-shot settings. This further validates our intriguing finding.
> ***
> ### Response to reject reason 6:
> > "The experiments are performed only on a small subset (4 datasets) of BEIR benchmark. It is not sufficient to support the claims made in this paper. Thorough evaluation on a larger set is important to support the paper's claims."
>
> Due to limited computational resources and numerous LLMs with various settings to run, and in order to ensure feasibility, we decided to choose the most widely used subset of BEIR that is used in most literature. Despite being a subset, it comprises a total of 1347 queries with deep ranking judgments across 4 distinct domains. We believe that it is sufficient to substantiate our claims.

---

### Official Review · Reviewer_1y9F · 2023-08-12

**Typos Grammar Style And Presentation Improvements:** Please refer to "Reasons To Reject".
**Soundness:** 4

**Excitement:**

4: Strong: This paper deepens the understanding of some phenomenon or lowers the barriers to an existing research direction.

**Missing References:**

Encourage the author to consider citing these related works that also use a similar approach to Eqn. (1) for text reranking.

Niklas Muennighoff. 2022. SGPT: GPT sentence embeddings for semantic search.

Percy Liang et al. 2022. Holistic evaluation of language models.

**Paper Topic And Main Contributions:**

This paper empirically evaluated the zero-shot text reranking performance of open-source Large Language Models (LLMs). The experiments demonstrate that a purely zero-shot LLM can even outperform an LLM that has undergone instruction-tuning in terms of text reranking performance. The experiments conducted in this paper are detailed and the conclusions are quite surprising.

**Questions For The Authors:**

N/A

**Reasons To Accept:**

* Timely research on LLMs' zero-shot text reranking performance.
* Extensive experiments are conducted on subset of BEIR.
* Code is publicly available for reproduction.
* Observations are supervising and insightful.
* Limitations are thoroughly discussed.

**Reasons To Reject:**

The several surprising experimental conclusions in Section 4 are scattered throughout the very lengthy plain text. It is suggested that the author(s) organize it more clearly, giving it more structure, so the readers can more easily access these conclusions.

**Reproducibility:**

5: Could easily reproduce the results.

**Reviewer Confidence:**

3: Pretty sure, but there's a chance I missed something. Although I have a good feel for this area in general, I did not carefully check the paper's details, e.g., the math, experimental design, or novelty.

---

> ### Author Rebuttal · Authors · 2023-08-28
>
> - Thank you for the suggestion on improving the structure of the presentation of the results. This is a good suggestion that we will adopt to improve the presentation of the paper.
> - Thank you for the additional references, which we will include in the paper

---

### Meta-Review · Area_Chair_mQEG · 2023-09-19

**Recommendation:** 2

**Metareview:**

This short paper offers a series of experiments with open-source LLM for document re-ranking. Some surprising results are obtained, such as instruction-tuning degrades the retrieval effectiveness. While the phenomenon is observed, the paper falls short in providing plausible explanations. There is thus limited insight about the observations.

The paper could be published for the surprising results obtained.

---

### Decision · Program_Chairs · 2023-10-07

**Decision:**

Accept-Findings

**Comment:**

This short paper offers a series of experiments with open-source LLM for document re-ranking. Some surprising results are obtained, such as instruction-tuning degrades the retrieval effectiveness. While the phenomenon is observed, the paper falls short in providing plausible explanations. There is thus limited insight about the observations.

The paper could be published for the surprising results obtained.